# Prognostic Role of Neutrophil-to-Lymphocyte Ratio (NLR), Lymphocyte-to-Monocyte Ratio (LMR), Platelet-to-Lymphocyte Ratio (PLR) and Lymphocyte-to-C Reactive Protein Ratio (LCR) in Patients with Hepatocellular Carcinoma (HCC) undergoing Chemoembolizations (TACE) of the Liver: The Unexplored Corner Linking Tumor Microenvironment, Biomarkers and Interventional Radiology

**DOI:** 10.3390/cancers15010257

**Published:** 2022-12-30

**Authors:** Roberto Minici, Maria Anna Siciliano, Michele Ammendola, Rita Carlotta Santoro, Vito Barbieri, Girolamo Ranieri, Domenico Laganà

**Affiliations:** 1Radiology Division, Pugliese-Ciaccio Hospital, 88100 Catanzaro, Italy; 2Oncology Division, Pugliese-Ciaccio Hospital, 88100 Catanzaro, Italy; maria.anna.siciliano@hotmail.com (M.A.S.); v.barbieri@materdominiaou.it (V.B.); 3Digestive Surgery Unit, Science of Health Department, Magna Graecia University, 88100 Catanzaro, Italy; michele.ammendola@unicz.it; 4Haemophilia and Thrombosis Center, Pugliese-Ciaccio Hospital, 88100 Catanzaro, Italy; ritacarlottasantoro@gmail.com; 5Interventional and Medical Oncology Unit, IRCCS Istituto Tumori “Giovanni Paolo II”, 70124 Bari, Italy; g.ranieri@oncologico.bari.it; 6Radiology Unit, Department of Experimental and Clinical Medicine, Magna Graecia University, 88100 Catanzaro, Italy; domenico.lagana@unicz.it

**Keywords:** inflammation-based scores, neutrophil-to-lymphocyte ratio (NLR), lymphocyte-to-monocyte ratio (LMR), platelet-to-lymphocyte ratio (PLR), lymphocyte-to-C reactive protein ratio (LCR), hepatocellular carcinoma (HCC), transcatheter arterial chemoembolization (TACE), tumor microenvironment, biomarkers, cancer

## Abstract

**Simple Summary:**

Response to TACE is a surrogate marker of tumor aggressive biology, with manifold practical implications. In turn, inflammation-based scores are biomarkers of the relationship between the tumor stromal microenvironment and the immune response. Investigating the connection among the tumor stromal microenvironment, biomarkers and the response to TACE is crucial to recognize TACE refractoriness/failure, thus providing patients with tailored therapeutics. This review aims to provide a comprehensive overview of the prognostic roles of the neutrophil-to-lymphocyte ratio (NLR), the lymphocyte-to-monocyte ratio (LMR), the platelet-to-lymphocyte ratio (PLR), and the lymphocyte-to-C reactive protein ratio (LCR) in patients with HCC undergoing chemoembolization of the liver. Inflammation-based scores may be convenient, easily obtained, low-cost, and reliable biomarkers with prognostic significance for HCC undergoing TACE. Further investigations with large cohorts of patients are required to establish the most appropriate cut-off values, thus consolidating the use of inflammation-based scores in clinical practice.

**Abstract:**

TACE plays a pivotal role in hepatocellular carcinoma, from disease control to downstaging and bridging to liver transplant. Response to TACE is a surrogate marker of tumor aggressive biology, with manifold practical implications such as survival, the need for more aggressive treatments in the intermediate stage, the selection of patients on the transplant waiting list, the dropout rate from the transplant list and the post-transplant recurrence rate. Inflammation-based scores are biomarkers of the relationship between the tumor stromal microenvironment and the immune response. Investigating the connection among the tumor stromal microenvironment, biomarkers, and the response to TACE is crucial to recognize TACE refractoriness/failure, thus providing patients with tailored therapeutics. This review aims to provide a comprehensive overview of the prognostic roles of the neutrophil-to-lymphocyte ratio (NLR), the lymphocyte-to-monocyte ratio (LMR), the platelet-to-lymphocyte ratio (PLR), and the lymphocyte-to-C reactive protein ratio (LCR) in patients with HCC undergoing chemoembolization of the liver. Inflammation-based scores may be convenient, easily obtained, low-cost, and reliable biomarkers with prognostic significance for HCC undergoing TACE. Baseline cut-off values differ between various studies, thus increasing confusion about using of inflammation-based scores in clinical practice. Further investigations should be conducted to establish the optimal cut-off values for inflammation-based scores, consolidating their use in clinical practice.

## 1. Introduction

Liver cancer is the seventh cancer in the number of new cases and the third most frequent cause of cancer-related death globally, with 905,677 new cases and 830,180 deaths per year [1].

Intermediate-stage hepatocellular carcinoma (HCC) or stage B disease, classified according to the Barcelona Clínic Liver Cancer (BCLC) staging system, has transcatheter arterial chemoembolization (TACE) as the first-choice treatment, according to the EASL Guideline for the management of HCC [2,3,4]. According to the 2022 BCLC strategy [4], HCC should not be treated with TACE only in case of diffuse, infiltrative, and extensive bilobar liver involvement. Furthermore, successful downstaging can also make patients initially outside the Milan criteria considered for liver transplantation (LT) [5]. Interestingly, the presence of favorable histological features (i.e., absence of microvascular invasion and low tumor grading) may be indicated by a good response to downstaging, similar to patients classified within the Milan criteria from the beginning.

Hence, tumor aggressiveness can be predicted by the response to downstaging [6,7]. Downstaging of HCC can be achieved through various loco-regional treatments, among which TACE is the most studied and used in clinical practice [8,9,10,11,12]. Moreover, locoregional treatments (LRTs) can play a pivotal role in preventing progression outside transplantation criteria, in patients with stage 0 or stage A HCC in the transplant list [13,14,15,16]. If there is an expected waiting time in the transplant list of more than 6 months according to AASLD guidelines or more than 3 months according to ESMO guidelines, bridging therapy with LRT is recommended [17,18]. TACE has been progressively recognized as the standard of treatment for delaying disease progression [19,20,21,22]. More in-depth, TACE is particularly useful in cases where a partial or complete response can be obtained. Therefore, the data guide scientists to consider the response to LRT as a surrogate marker of unfavorable tumor biology [23,24,25].

Inflammation-based scores, such as the neutrophil-to-lymphocyte ratio (NLR), the lymphocyte-to-monocyte ratio (LMR), the platelet-to-lymphocyte ratio (PLR), and the lymphocyte-to-C reactive protein ratio (LCR), depend on circulating neutrophils, lymphocytes, platelets, monocyte and C-reactive protein (CRP) counts. Neutrophils enhance angiogenesis, the proliferation of tumor cells, metastasis, and their evasion of the immune response [26]. Platelets facilitate tumor progression by supporting cancer stem cells, inducing angiogenesis, sustaining cell proliferation, and evading immune surveillance [27]. In addition, a worse cell-mediated immune response to cancer may be associated with relative lymphocytopenia [28]. Serum CRP reflects systemic inflammatory response and, if combined with other markers of the systemic inflammatory response, it is a useful prognostic biomarker for several tumor types [29]. Tumor progression and escape from the immune response are enhanced by the aforementioned factors. Hence, an HCC-promoting state can be suspected from high levels of inflammation-based scores, thus resulting in a poor prognosis after TACE [30].

Therefore, TACE is performed in HCC patients in a wide range of settings, such as local tumor control, downstaging, and bridging to LT. Increasing evidence demonstrates that response to TACE is a surrogate marker of tumor aggressive biology in patients with very early-, early-, and intermediate-stage HCC. In turn, inflammation-based scores are biomarkers of the relationship between the tumor stromal microenvironment and the immune response, and they have been proposed as prognostic factors to predict the recurrence, disease progression, and survival of patients with HCC, as well as predictive factors of the patient’s response to TACE [8,13,31,32,33]. Investigating the connection among the tumor stromal microenvironment, biomarkers, and the response to TACE is crucial to recognize TACE refractoriness/failure, thus providing patients with tailored therapeutics.

This narrative review aims to provide a comprehensive overview of the prognostic roles of the neutrophil-to-lymphocyte ratio (NLR), the lymphocyte-to-monocyte ratio (LMR), the platelet-to-lymphocyte ratio (PLR), and the lymphocyte-to-C reactive protein ratio (LCR) in patients with HCC undergoing chemoembolization of the liver.

## 2. Data Collection Strategy

Online databases (PubMed and Google Scholar) were searched for peer-reviewed articles (including case reports and letters to editors) in which the prognostic role of scores in patients with HCC undergoing chemoembolization was analyzed; articles published from January 2000 to August 2022 were included. The search period was expanded to the timeframe 1980–2022 to include articles investigating the role of immune response in cancer and that of the tumour stromal microenvironment. Searches were conducted using relevant keywords such as “inflammation-based scores”, “neutrophil-to-lymphocyte ratio”, “NLR”, “lymphocyte-to-monocyte ratio”, “LMR”, “platelet-to-lymphocyte ratio”, “PLR”, “lymphocyte-to-C reactive protein ratio”, “LCR”, “HCC”, “liver cancer”, “chemoembolization”. These key terms were used in multiple combinations to create strings to search study records’ titles and abstracts. Investigations unrelated to the prognostic role of inflammation-based scores in patients with HCC undergoing liver chemoembolization were excluded. Table 1 summarizes all relevant studies included in the literature review that evaluated the prognostic role of preoperative inflammation-based scores in patients undergoing TACE. In addition, the statistical methods used to evaluate the prognostic role have been reported, and where specified in the various studies, the statistical tests used to select the specific cut-off values have been indicated in parentheses.

## 3. Neutrophil-to-Lymphocyte Ratio (NLR)

Different diseases, including infectious diseases, metabolic diseases, autoimmune diseases, ageing-associated diseases and cancer, promote pre-neutrophils expansion and immature neutrophils release from bone marrow [34]. Although neutrophils exert a positive action in the host defense [35], they are involved in various mechanisms resulting in pathogenesis. Proteases and reactive oxygen species (ROS) released by neutrophils in the target site determine tissue damage and chronic inflammation [36]. Secondly, neutrophils may cause immunosuppression by lowering response to chemokines and by inhibiting T cell immunity [37]. Finally, neutrophil extracellular traps (NETs) extruded by activated neutrophils are responsible for atherosclerotic plaque destabilization, induction of thrombosis and anti-self-antibodies production [38,39,40]. In recent years, the cancer-promoting effects of neutrophils have gained increasing attention. Neutrophils exert a key role in cancer carcinogenesis; they promote cancer initiation by enhancing inflammation pathways, induce DNA damage by producing genotoxic DNA substances and promote neoangiogenesis and immunosuppression [41]. More thoroughly, NETs foster inflammation in patients with nonalcoholic steatohepatitis (NASH) promoting HCC development; deoxyribonuclease treatment or peptidyl arginine deaminase type IV knockout inhibits HCC by decreasing NET formation [42]. In addition, NETs facilitate naïve CD4+ T cells metabolic reprogramming, resulting in a positive correlation with the number of regulatory T cells (Tregs) in cancer. Cancer immunosurveillance is promoted by therapies targeting the interaction between NETs and naïve CD4+ T cells or inhibiting Treg activity, thus preventing HCC formation [43]. Neutrophils play a pivotal role also in cancer progression, in cancer metastases by promoting cancer cell migration, intravasation, and extravasation and forming premetastatic niches and in cancer recurrence [41,44]. The tumor microenvironment (TME) consists of non-cancerous cells, such as T cells, adipocytes, neutrophils, macrophages, and stromal cells. In cancer, the presence of different subpopulations of neutrophils explains the coexistence of pro-cancer and anti-cancer effects exerted by neutrophils. Various cytokines secreted by cancer cells can modify the cancer microenvironment by reprogramming neutrophils and converting them to a cancer-promoting or antitumor polarity [41,45]. In a genetically engineered mouse model (GEMM) of lung adenocarcinoma, transforming growth factor-β (TGFβ) activation or blockade can polarize neutrophils in a protumor or an antitumor phenotype, named N2 and N1, respectively [46]. 

Meanwhile, relative lymphocytopenia may reflect a poorer cell-mediated immune response to cancer [28]. Tumor-associated antigens can be recognized by tumor-infiltrating T lymphocytes, which can trigger anti-tumor immune response [47]. Cancerous cells reduce cytotoxic T lymphocyte (CTL) proliferation in the tumor by producing immunosuppressive cytokines such as interleukin (IL)-10, vascular endothelial growth factor (VEGF), and TGF- β and by consuming IL-2, a cytokine that plays a key role in maintaining CTL function [48]. While high lymphocyte counts predict a better prognosis and improved survival in HCC, low lymphocyte counts are associated with poor clinical outcomes [49,50,51].

Hence, the NLR is influenced by an immune microenvironment that promotes vascular invasion by tumor and hinders the host’s immune surveillance. In clinical practice, the NLR has been proposed as an independent prognostic indicator for patients with cancer [52], as well as a marker of the systemic inflammatory response [53]. The host’s inflammatory reaction to cancer and/or the systemic effects determined by the cancer cells leads to the upregulation of the inflammatory response [54,55]. Increased systemic inflammation, detectable by the NLR, correlates with cancer progression, metastasis, and clinical outcome in a variety of tumors [56,57,58,59,60,61,62,63].

In a retrospective evaluation of 605 consecutive patients with HCC initially treated by TACE, the preoperative NLR was an independent risk factor for disease progression and overall survival (OS) in the multivariate analysis. A time-dependent receiver operating characteristic (ROC) analysis for predicting progression within 1 year showed a cut-off value of 1.7 for the NLR. The Kaplan−Meier analysis demonstrated that the group with an NLR of ≥ 1.7 exhibited a shorter time to progression (TTP), as well as decreased OS [30]. In a retrospective study including 46 patients with treatment-naïve HCC who received drug-eluting beads (DEB)-TACE, a higher baseline NLR was predictive of a poorer tumor response (volumetric enhancement-based tumor response) and a shorter progression-free survival (PFS) after DEB-TACE [64]. In a retrospective evaluation of treatment-naïve patients who received TACE as first-line treatment for stage B HCC, multivariable logistic regression analysis found that disease progression 6 months after TACE can be predicted by a baseline NLR of ≥ 3 [65]. Recently, in a cohort of 380 consecutive patients newly diagnosed with HCC and treated with TACE, a significant survival difference was found between a group with a normal baseline NLR (≤2.4) and a group with an increased baseline NLR (>2.4), with median OS values of 29.1 and 19.1 months, respectively [66]. Other investigations have highlighted that the baseline NLR should be considered a good prognostic factor of survival [67,68,69] and disease progression [67,70] in HCC patients undergoing TACE. These findings were confirmed by the results of a recent meta-analysis by Li et al., which demonstrated that a high preoperative NLR is associated with a poor OS in HCC patients treated by TACE [71]. Similarly, dynamic changes after embolization relative to the baseline NLR are also considered independent predictors of survival [72]. Lastly, a metanalysis by Xiao et al. showed that a high preoperative NLR was significantly associated with a poor OS of HCC patients initially treated by TACE, as well as it was significantly correlated with the presence of vascular invasion, multiple tumors (satellite nodule) and high levels of serum alpha-fetoprotein (AFP) (≥400 ng/mL) [31]. More thoroughly, Xiao et al. evaluated the prognostic function of different NLR cut-off values (1.9, 2–3, 3–4, 4, and 5) reported by the studies included in their meta-analysis [31]. They have found that many NLR cut-off values (1.9, 3–4, 4, and 5) were statistically correlated with poor OS of HCC and that an NLR of 5 was the most commonly used in the analyzed studies. Therefore, a high NLR is correlated with a more aggressive phenotype of HCC which is in turn closely associated with poor OS and dismal outcomes [31].

Therefore, the NLR is influenced by the TME and may be a convenient, easily-obtained, low-cost, and reliable biomarker with prognostic significance for HCC undergoing TACE [31]. However, various studies differ in the NLR cut-off values used and in the different subpopulations of HCC analyzed [31], thus leading to between-study heterogeneity and increasing confusion about the use of the NLR in clinical practice. Further investigations with large cohorts of patients are desirable to better define the most appropriate NLR cut-off value.

## 4. Lymphocyte-to-Monocyte Ratio (LMR)

The LMR has been recently evaluated as a prognostic marker in patients with solid tumors [73,74], including patients with HCC [75,76]. However, the mechanism by which the LMR affects the prognosis of tumor patients has not been fully understood [77]. 

Lymphocytes play a pivotal role in immunosurveillance and immune editing, and lymphocyte infiltration in the TME contributes to the immunologic anticancer reaction [78,79]. The presence of tumor-infiltrating lymphocytes is associated with improved survival in various cancers, and conversely, low lymphocyte counts and failure to infiltrate the tumor lead to inferior survival [80,81]. Tumor-associated macrophages (TAMs) are a subpopulation of monocytes that originates in the circulating blood and is activated around the tumor due to the release of tumor chemokines [82]. TAMs produce growth factors and cytokines, thus influencing, directly and indirectly, the metastatic process of tumor cells by modulating the tumor microenvironment [82]. The correlation between high TAM infiltration and poor prognosis has been confirmed by various studies [77,82,83]. The evaluation of the TAMS can be carried out through the monocytes of the peripheral blood, which act as a biological marker. Therefore, the LMR can be used as a biomarker of both lymphocytes and monocytes and as a prognostic factor for patients with cancer [77]. A recent investigation highlighted that the survival of cancer patients is improved by the inhibition of the programmed death (PD) 1/programmed death-ligand (PD-L) 1 immune checkpoint pathway [84]. PD-1 lies over the plasma membrane of T lymphocytes, and T-cell activity is reduced by the interaction between PD-1 and PD-L1 or PD-L2, thus downregulating the immune response against cancer. PD-1- or PD-L1-blocking antibodies inhibit these interactions, resulting in the suppression of cancer immune escape and inducing a T cell-mediated response [85]. A statistically significant association between the low LMR and the expression of PD-L1 was reported in patients with HCC, hypothesizing that increased levels of monocyte-derived cells in the HCC microenvironment can be suspected if a low LMR is detected, resulting in an increased PD-L1 expression level of HCC cells in response to cytokines secreted by the immune cells [86]. The association between TME immune checkpoint pathways and the low LMR has also implications in clinical practice. Indeed, a high LMR without significant PD-L1 expression was found to be a prognostic factor for improved OS and recurrence-free survival (RFS) in patients with HCC following hepatic resection [86]. Moreover, the LMR may be considered an early indicator for predicting the outcome of patients receiving anti-PD-1 agents [77].

Various investigations have shown that a decreased pretreatment LMR has an unfavorable impact on OS in cancer patients among various tumor subgroups [87], including HCC [75,76,88]. A more aggressive tumor behavior, including a larger tumor size and a higher serum AFP concentration, is demonstrated in HCC patients with a low preoperative LMR when compared with in patients presenting a high preoperative LMR [86]. In a mixed cohort of 204 patients with HCC who underwent radiofrequency ablation (RFA) and TACE, patients with a higher LMR had a longer OS than those with a lower LMR; in addition, the Cox proportional-hazards model identified the LMR as a prognostic factor for OS [89]. In a retrospective evaluation of 180 patients, a combination of a baseline high NLR plus a low LMR was effective for predicting OS in HCC after TACE [90]. A higher monocyte-to-lymphocyte ratio is associated with a statistically significant poorer RFS and OS in patients with HCC who were treated with TACE combined with local ablation [91]. In a retrospective study evaluating 128 intermediate-stage HCC patients who received TACE alone as an initial treatment, a statistically significant difference in PFS was found between the group with a high LMR and the group with a low LMR (LMR cut-off value: 4.4) [92]. In a recent investigation involving patients with intermediate-stage HCC, a combination of an LMR (>4)/an NLR (<7.2) was found to be an independent prognostic factor for successful downstaging after DSM-TACE [8]. In patients with early-stage HCC and Child-Pugh stage B undergoing TACE as a bridging therapy while awaiting LT, the subgroup with an LMR of >4 and an NLR of <7.2 showed a better time to dropout from the transplant waiting list. In addition, a combination of an LMR of >4 and an NLR of <7.2 was found to be an independent prognostic factor for dropout from the transplant waiting list [13]. To the best of our knowledge, to date, no meta-analytic investigations were conducted to investigate the baseline LMR as a prognostic factor in patients with HCC undergoing TACE.

Hence, the preoperative LMR may represent the balance between antitumor immune reaction and tumor promotion function [93], and as an indicator of the inflammatory response, it has the characteristics of simple, fast, operable, specific and sensitive detection methods, endowed with prognostic relevance in HCC, including those patients undergoing TACE. In a recent meta-analysis, it has been shown that the LMR cut-off level varied and ranged from 1.1 to 5.26, and although some studies reported that the receiver operating characteristic curves (ROCs) were used to determine the cut-offs, the approach to choosing LMR cut-offs remains unclear in many papers [87]. Koh et al. identified the LMR as being related to the age of patients [94]. Therefore, further large prospective studies should be conducted to establish the optimal LMR cut-off, based on the type of tumor and the patient’s age. 

## 5. Platelet-to-Lymphocyte Ratio (PLR)

The PLR has been identified as a negative prognostic factor in several advanced tumors [95]. The prognostic value of the PLR in HCC has also been investigated. However, it remains unclear how this ratio can improve disease progression and OS [49,67]. Lai et al. have performed a meta-analysis demonstrating that high pre-LT PLR values correspond to a 3.33-fold increased risk of HCC recurrence, even in a multivariable model comprehending features of tumor biology and morphology [96]. The PLR, as well as the NLR, independently predicts HCC recurrence and survival [97]. 

In addition to their classic role as effector cells in hemostasis, platelets have been reported to have a pro-tumoral role by protecting tumor cells from natural-killer cells’ lysis and promoting metastasis [98]. An elevated PLR indicates an activated inflammation response, a stimulating transcription factor as nuclear transcription factor-kappa B (NF-kB), a signal transducer and an activator of transcription 3 (STAT3), hypoxia-inducible factor 1-alpha (HIF-1A), and pro-inflammatory cytokines including tumor necrosis factor (TNF)-alpha, IL1b, and IL-6. These proteins play a critical role in tumor-cell proliferation, survival, migration, and invasion, promoting epithelial−mesenchymal transition (EMT), as well as angiogenesis, metastatization, and the response to chemotherapy [99,100,101]. Therefore, platelets might represent a potential therapeutic target, and indeed, sorafenib is a standard of care in HCC treatment directly targeting pathways mediated by VEGF and platelet-derived growth factor (PDGF) [102]. Moreover, previous investigations highlighted the inhibiting effect on tumor growth exerted by antiplatelet therapies (aspirin, warfarin, and cyclo-oxygenase inhibitors), even in HCC [103,104]. 

Regarding the predictive role of the PLR in patients receiving TACE for HCC, Xue et al. have demonstrated that the PLR can be used to predict poor survival in patients that received TACE for HCC. A PLR value of 150 might be used as a cut-off value in predicting the outcomes of HCC patients undergoing TACE. Moreover, a baseline PLR of >150 calculated before the first chemoembolization in huge HCC (diameter exceeding 10 cm) can predict OS. The 12-, 24-, and 36-month survival rates in the high PLR group (22.6%, 8.1%, and 4.1%, respectively) were significantly lower than those in the low PLR group (35.6%, 22.4%, and 14%, respectively) [49]. Interestingly, Lai et al. have reported the same PLR cut-off value for patients undergoing LT, and the sub-analysis of patients treated with TACE has indicated that a PLR of >150 can stratify patients according to the risk of post-transplantation recurrence [96]. Conversely, inflammatory biomarkers might not have a predictive role in smaller tumors, which are probably less biologically active [105]. Interestingly, He et al. demonstrated that the NLR−PLR score negatively affect OS in HCC patients treated with TACE, with better performance than the NLR or PLR alone [69].

A retrospective trial has enrolled HCC patients receiving DEB-TACE to evaluate if inflammatory markers, such as the PLR, could correlate with tumor response and radiomic features extracted from pretreatment contrast-enhanced magnetic resonance imaging. Interestingly, a higher baseline PLR were predictive of a poorer tumor response (*p* = 0.004) and a shorter PFS (*p* < 0.001) after DEB-TACE. When compared to baseline imaging, a high PLR correlated with non-nodular tumor growth (*p* < 0.001) [64].

Several studies have shown that a high PLR correlates with poor outcomes in HCC patients receiving TACE in combination with systemic therapy. The anti-PD-L1 atezolizumab plus bevacizumab have been recently approved for first-line treatment of advanced/metastatic HCC [106]. Wang et al. have performed a biomarker analysis demonstrating that the NLR and the PLR are able to predict PFS in a cohort of patients receiving atezolizumab plus bevacizumab. The NLR and the PLR are independent prognostic factors for PFS in univariate/multivariate analysis (PLR cut-off value of 230) [107]. Liu et al. performed a meta-analysis demonstrating that patients receiving sorafenib with a lower baseline PLR have a better OS [108]. The PLR could be used to predict OS in patients with HCC who receive first-line Lenvatinib [109]. Furthermore, the PLR has been reported to predict response in patients with HCC receiving anti-PD-1 therapy, such as nivolumab [110]. In addition, apatinib monotherapy improves OS in pretreated advanced HCC compared to the placebo, with an acceptable safety profile [111]. Of interest, a recent clinical trial has investigated the PLR value in patients receiving a combination of TACE and the VEGFR-inhibitor apatinib [112]. The OS and the TTP in patients receiving TACE plus apatinib have been compared between two groups (PLR > 150 and PLR ≤ 150), demonstrating poorer outcomes when the PLR was greater than 150 [112]. In the multivariate analysis, only the PLR value was identified as an independent prognostic factor [112]. No significant differences are highlighted comparing TACE plus apatinib and TACE alone in the subgroups with a PLR of >150 [112]. A cut-off value of 150 for the PLR could be used to predict the outcomes of advanced HCC patients receiving TACE plus apatinib [112]. Notably, HCC patients with a PLR value of >150 might not be suitable for TACE−apatinib treatment [112]. 

In conclusion, high PLR in patients with HCC candidates to TACE could represent a powerful, inexpensive, widely obtained, repeatable, and reliable predictive biomarker of response and survival. More studies are warranted to draw stronger conclusions that may be significant for clinical practice.

## 6. Lymphocyte-to-C Reactive Protein Ratio (LCR)

Blood CRP is a protein that can be easily measured from blood samples using qualitative, semi-quantitative, and quantitative approaches [113]. This protein is synthesized from healthy hepatocytes and HCC cells, and it is secreted into the plasma as a pentamer [114]. Of note, two concentration thresholds were established: conventional CRP levels (≥10 µg/mL) and high sensitivity CRP levels (hsCRP; <10 µg/mL) [115]. Moreover, the significance of hsCRP levels in cancer is under investigation. As an unspecific marker of inflammation, caution is necessary for its interpretation, which must always be correlated to the clinical evaluation, as per FDA guidance [113]. A critical interpretive issue needs to be considered: CRP is a dynamic biomarker, and its blood level can change rapidly according not only to disease severity, but also to other clinical factors, which should be considered to avoid improper patient management [113]. Activated endothelial cells allow platelets, neutrophils, and blood proteins (e.g., IL-6, IL1b, and CRP) to enter tissue to activate an anti-tumor response [115]. The proinflammatory cytokine IL6, usually highly expressed in the TME, is the principal regulator of CRP, and it is related to HCC progression and metastasis [116]. Cells activated by monomeric CRP can stimulate intracellular signaling pathways, including the activation of the NF-kB transcription factor [117]. CRP directly binds fibronectin stimulating the retention of monocytes in the TME [118]. Moreover, CRP can enhance cytotoxic T lymphocyte-mediated cell lysis and, on the other hand, can promote a sustained pro-inflammatory and pro-tumor immune microenvironment. Hence, the role of CRP has been recognized in the acute phase response and chronic inflammation, too [117]. CRP has been identified as a diagnostic tool in assessing disease status and progression even in cancer [115]. Moreover, an elevated-serum CRP is closely related to the risk of malnutrition and neoplastic cachexia, leading to increased mortality and reduced effectiveness of therapies in patients with cancer [119]. The CRP/albumin ratio (CAR) represents a prognostic inflammatory marker in HCC patients, and it could predict recurrence after therapy [120]. Moreover, a correlation has been demonstrated combining a high CAR and a high NLR with OS, confirming its role in malnutrition in this cohort of patients.

The role of inflammatory markers in predicting outcomes in cancer patients was highlighted using several scoring systems. The Glasgow Prognostic Score, the modified Glasgow Prognostic Score, and the high-sensitivity-modified Glasgow Prognostic Score represent independent prognostic factors in HCC and other neoplasms by measuring serum CRP and albumin levels [121,122].

Carr et al. have demonstrated a significant association between blood CRP levels, parameters of HCC aggressiveness and blood AFP levels, suggesting a role in HCC growth and invasion [123]. Moreover, Zheng et al. performed a systematic review and meta-analysis demonstrating a correlation between high-serum CRP and poor OS and RFS in HCC patients. No differences emerged based on the treatment (surgical vs. non-surgical), the region (east vs. west), the measurement of CRP (CRP vs. hsCRP), and the maximal follow-up period (more than 5 years vs. less than 5 years) [124]. A statistically significant correlation was also demonstrated between high-serum CRP and the presence of vascular invasion, multiple tumors, tumor size, and the TNM stage [124]. In the immunotherapy era, a recent manuscript has investigated the prognostic value of the baseline CRP in HCC patients treated with PD-1 inhibitors [125].

The LCR has been proposed as a predictive and prognostic marker of outcomes in cancer patients, including HCC [126,127,128,129]. 

Interestingly, the utility of the LCR score in HCC patients who undergo TACE has not been fully defined. Lu et al. evaluated the LCR score, as well as several inflammation-based scores in predicting OS for HCC patients who underwent TACE [129]. One-thousand six-hundred and twenty-five patients were enrolled in the aforementioned study, and 55.7% had an LCR score of >6000. The 1-year and 3-year OS rates were 71.9% and 32.4%, respectively, in patients with an LCR of >6000 compared to 40.1% and 13.7%, respectively, in patients with an LCR of <6000 (*p* < 0.001). Thus, the LCR is an independent prognostic factor for improved OS in this subset of patients (HR = 1.45, *p* < 0.001), except for patients with extrahepatic metastasis. Moreover, the LCR score is discriminative in risk stratification across different subgroups of HCC patients who exhibit different liver functions and tumor characteristics [129]. This is relevant in predicting patients who might benefit from TACE or other locoregional treatments. 

## 7. Conclusions

Inflammation-based scores are biomarkers of the relationship between the tumor stromal microenvironment and the immune response, and they have been proposed as convenient, easily-obtained, low-cost, and reliable biomarkers, endowed with prognostic relevance in HCC patients undergoing TACE. Investigating the connection among the tumor stromal microenvironment, biomarkers, and the response to TACE is crucial to recognize early TACE refractoriness/failure, thus providing patients with tailored therapeutics. Practical implications are manifold and include survival, the need for more aggressive treatments in the intermediate stage, the selection of patients on the transplant waiting list, the dropout rate from the transplant list, and the post-transplant recurrence rate. Further investigations should be conducted to establish the optimal cut-off values for each score and stage of HCC, thus consolidating the use of inflammation-based scores in clinical practice.

## Figures and Tables

**Table 1 cancers-15-00257-t001:** Summary table of the main studies included in the literature review that evaluated the prognostic role of preoperative inflammation-based scores.

Marker	Reference	Study Design	Sample Size	Treatment	Cut-off value	*p*-Value	Statistical Method	Endpoint
NLR	Li (2020)	Metanalysis	4023	TACE	5 2.5–5 2.5	<0.00001 <0.00001 <0.00001	Random Effect Fixed Effect Fixed Effect	OS
NLR	Xiao (2014)	Metanalysis	302	TACE	NA	<0.0001	Fixed Effect	OS
NLR	Cho (2022)	Retrospective	605	cTACE	1.7 1.7	0.007 <0.001	Cox (ROC) Cox (ROC)	TTP OS
NLR	Chu (2021)	Retrospective	495	cTACE	3 3	0.007 <0.001	Cox (ROC) Logistic Regr.	OS 6m-PD *
NLR	Wang (2020)	Retrospective	380	cTACE	2.4	0.027	Cox (Median)	OS
NLR	Liu (2020)	Retrospective	180	cTACE	3.94	<0.001	Cox (ROC)	OS
NLR	Schobert (2020)	Retrospective	46	DEB-TACE	NA 3.22	0.014 0.002	Linear Regr. Cox (Mean)	ETV ** PFS
NLR	He (2019)	Retrospective	216	cTACE	1.77	<0.0001	Log-rank	OS
NLR	Rebonato (2017)	Retrospective	72	DEB-TACE	2.03	0.0028	Cox (median)	OS
NLR	Fan (2015)	Retrospective	132	cTACE	3.1	0.130	Cox (mean)	OS
LMR	Wang (2022)	Prospective	606	cTACE + ablation	2.27 ^†^ 2.27 ^†^ 2.27 ^†^	0.022 0.029 0.011	Cox (Youden) Cox (Youden) Logistic Regr.	RFS OS 2y-Recurrence
LMR	Liu (2021)	Retrospective	128	TACE	4.4	0.236	Cox (ROC)	PFS
LMR	Liu (2020)	Retrospective	180	cTACE	2.2	<0.001	Cox (ROC)	OS
LMR	Shen (2019)	Retrospective	204	cTACE + RFA	2.13	<0.0001	Cox (ROC)	OS
PLR	Li (2020)	Metanalysis	856	TACE	NA	0.007	Random effect	OS
PLR	Liu (2021)	Retrospective	128	TACE	92	0.000195	Cox (ROC)	PFS
PLR	Chen (2020)	Retrospective	134	cTACE + apatinib	150	0.014	Cox	OS
PLR	Schobert (2020)	Retrospective	46	DEB TACE	NA 113.1	0.004 <0.001	Linear Regr. Cox (mean)	ETV * PFS
PLR	He (2019)	Retrospective	216	cTACE	94.62	0.0022	Log-rank	OS
PLR	Shen (2019)	Retrospective	204	cTACE + RFA	95.65	<0.0001	Cox (ROC)	OS
PLR	Xue (2015)	Retrospective	291	cTACE	150	0.002	Cox (chi-square)	OS
PLR	Fan (2015)	Retrospective	132	cTACE	137	<0.001	Cox (mean)	OS
LCR	Lu (2021)	Retrospective	1625	cTACE	6000	<0.001	Cox (ROC)	OS

* 6m-PD—progressive disease at 6 months; ** ETV—enhancing tumor volume; ^†^—calculated with the formula 1/MLR.

## Data Availability

Not applicable.

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
