# Peer review of "Prognostic Role of Neutrophil-to-Lymphocyte Ratio (NLR), Lymphocyte-to-Monocyte Ratio (LMR), Platelet-to-Lymphocyte Ratio (PLR) and Lymphocyte-to-C Reactive Protein Ratio (LCR) in Patients with Hepatocellular Carcinoma (HCC) undergoing Chemoembolizations (TACE) of the Liver: The Unexplored Corner Linking Tumor Microenvironment, Biomarkers and Interventional Radiology"

_cancers, 2022, doi:10.3390/cancers15010257_

Round 1

Reviewer 1 Report

In this review, Minici and colleagues aimed to provide a comprehensive overview of the prognostic role of neutrophil-to-lymphocyte ratio (NLR), lymphocyte-to-monocyte ratio (LMR), platelet-to-lymphocyte ratio (PLR) and lymphocyte to-C reactive protein ratio (LCR) in patients with HCC undergoing chemoembolizations of the liver. Despite the interest topic and clinical relevance of this work, there are some important aspects to consider in the writing.

1.    Once defined for the first time, only the abbreviation should be used, i.e., HCC lines 60 and 64; the same applies to NLR, MLR, PLR, LCR, TGFB, OS and others in the whole manuscript; please revise them and replace the full name by the proper abbreviation.

2.    There are many abbreviations without definition, i.e., IL, DEB-TACE, AFP, VEGF, PDGF, TME, AFP and others; please revise them.

3.    References are lacking in many sentences in the whole manuscript, please revise them. Some examples below:

“Inflammation-based scores, such as neutrophil-to-lymphocyte ratio (NLR), lymphocyte-to-monocyte ratio (LMR), platelet-to-lymphocyte ratio (PLR) and lymphocyte-to-C reactive protein ratio (LCR), could act as biomarkers of the relationship between host inflammatory response, involved in tumour-related angiogenesis, and immune system, responsible for cytotoxic cancer cells death.”

“The practical implications are manifold and concern, as well as survival, the need for more aggressive treatments in the intermediate stage, the selection of patients on the transplant waiting list, the drop-out rate from the transplant list and the post-transplant recurrence rate.”

“Lymphocytes play a pivotal role in immunosurveillance and immune-editing, and  lymphocyte infiltration in the tumor microenvironment is a prerequisite to an immunologic anticancer reaction.”

Tumor-associated macrophages (TAMs) are a subpopulation of monocytes that originates in the circulating blood and is activated around the tumor due to the release of tumor chemokines.

“Moreover, the significance of hsCRP levels in cancer is under investigation. As an unspecific marker of inflammation, caution is necessary for its interpretation, which must always be correlated to the clinical evaluation, as per FDA guidance. A critical interpretive issue needs to be considered: CRP level is a dynamic biomarker and its blood level can change rapidly according to disease severity but also to disease complications. Activated endothelial cells allow platelets, neutrophils and blood proteins (e.g. IL-6, IL1b, CRP) to enter tissue to activate an anti-tumor response.”

4.    It is not appropriate to compare the different cut-off points in different situations and populations. The authors could better explain this sentence. “More thoroughly, many NLR cut-off values (1.9, 4, 5) were statistically correlated with poor OS, although an NLR of 5 was the most commonly used in the analyzed studies.”

“CRP has been identified as a validated diagnostic tool in assessing cancer-related inflammation, disease status and progression in several tumors, including HCC.”

5.    To improve readers' understanding, the authors could present a summary table comparing the main studies of each marker, answering key questions such as the cut-off point used and how it was defined, sample size, result found.

6.    Please, revise the conclusion topic to highlight the answer of the study question. 

Author Response

Dear Reviewer,

Thanks for your comments and suggestions. We welcome them fully and have attempted to modify the manuscript to address your comments.

Here are our answers:

  1. The proper abbreviations have replaced the full names except in the study aim declaration and the conclusion.
  2. Missing abbreviations and definitions have been added.
  3. Fixed.
  4. The first sentence was a bit confusing, therefore the concepts have been reformulated (lines 205-215). The second sentence has been changed and the reference has been added.
  5. A summary table has been created and included. Both reviewers suggested adding a summary table, so it was created with suggestions from both.
  6. Conclusions have been reformulated.

Furthermore, editing of the English language and style has been performed.

Kind regards,

Roberto Minici

Reviewer 2 Report

In this review, the authors focused on the prognostic role of several inflammation-based scores in patients treated with transarterial chemoembolizations. The topic is interesting and highly debated; the physiopathological relationship between tumour stromal microenvironment and immune response is well described. However, this manuscript suffers of some limitations that can be addressed; in particular:

1.       The authors define this paper as a “comprehensive overview” of the prognostic role of Inflammation-based scores, however is not clear what was the strategy adopted for literature search. Were the Preferred Reporting Items for Systematic Reviews and Meta-analysis (PRISMA) or others criteria adopted? Even if the authors’ purpose was a simple narrative review, they should state the selection criteria for the studies described in this paper (relevance of publication, publication date, methodology…)

2.       Line 232. Only 1 out of 56 studies included in the Gu’s meta-analysis involved patients affected by hepatocellular carcinoma treated with hepatic resection (and not by TACE); thus, citing this evidence appears off topic with respect to this review dealing with the prognostic role of IBS in the specific setting of TACE-treated patients. 

3.       Regarding the most studied biomarker of systemic inflammation (NLR), some relevant clinical experiences in the specific clinical setting of HCC patients treated by TACE are lacking in this review; in particular:

a.        Pinato DJ, Sharma R. An inflammation‐based prognostic index predicts survival advantage after transarterial chemoembolization in hepatocellular carcinoma. Transl Res. 2012;160(2):146‐152. https://doi.org/10.1016/j.trsl.2012.01.011

b.        Rebonato A, Graziosi L, Maiettini D, et al. Inflammatory markers as prognostic factors of survival in patients affected by hepatocellular carcinoma undergoing transarterial chemoembolization. Gastroenterol Res Pract. 2017;2017:1‐9. https ://doi.org/10.1155/2017/4164130

c.        Taussig MD, Irene Koran ME, Mouli SK, et al. Neutrophil to lymphocyte ratio predicts disease progression following intra‐arterial therapy of hepatocellular carcinoma. HPB. 2017;19(5):458‐464. https ://doi.org/10.1016/j.hpb.2017.01.013

I encourage the authors to add these findings and extend the bibliographic search in order to provide a more detailed review on the prognostic usefulness of these markers. In particular, the results of the last meta-analysis published on this topic should be commented:

Li S, Feng X, Cao G, Wang Q, Wang L (2020) Prognostic significance of inflammatory indices in hepatocellular carcinoma treated with transarterial chemoembolization: A systematic review and meta-analysis. PLoS ONE 15(3):e0230879. https://doi.org/10.1371/journal.pone.0230879

4.       To summarize the evidences cited by authors and improve the appeal of this paper, I suggest providing a table for each of the IBS discussed (NLR, NLR, PLR, and LCR). Only papers dealing with the prognostic role of IBS in the specific clinical setting of TACE-treated patients should be included in the tables. The authors should describe the methodology of the studies (retrospective, case-control studies, meta-analysis…), the number of patients enrolled, the endpoints of the studies (PFS, OS…), witch cut-off was used and why and the statistical method used to investigate the prognostic role of each marker.

Author Response

Dear Reviewer,

Thanks for your comments and suggestions. We welcome them fully and have attempted to modify the manuscript to address your comments.

Here are our answers:

  1. The article has been marked as a narrative review and a short section on data collection strategy has been included.
  2. The meta-analysis by Gu et al was included to introduce the prognostic role of LMR in cancer broadly; soon after, specific evidence related to hepatocellular carcinoma followed. However, we understand that the sentence could generate misunderstandings, therefore we have reworked it, emphasizing which studies have specifically evaluated hepatocellular carcinoma and which have not.
  3. We added the relevant studies you suggested and commented on the main findings.
  4. A summary table has been created and included. Both reviewers suggested adding a summary table, so it was made with suggestions from both.

Kind regards,

Roberto Minici